# The Physiology of Postharvest Tea (*Camellia sinensis*) Leaves, According to Metabolic Phenotypes and Gene Expression Analysis

**DOI:** 10.3390/molecules27051708

**Published:** 2022-03-05

**Authors:** Shuang Mei, Zizi Yu, Jiahao Chen, Peng Zheng, Binmei Sun, Jiaming Guo, Shaoqun Liu

**Affiliations:** 1College of Engineering, South China Agricultural University, Guangzhou 510642, China; meishuang@gdaas.cn; 2Guangdong Academy of Agricultural Sciences, Guangzhou 510640, China; 3College of Horticulture, South China Agricultural University, Guangzhou 510642, China; yzz_123@stu.scau.edu.cn (Z.Y.); cjhtea@stu.scau.edu.cn (J.C.); zhengp@scau.edu.cn (P.Z.); binmei@scau.cn (B.S.); 4Maoming Branch, Guangdong Laboratory for Lingnan Modern Agriculture, Guangzhou 525000, China

**Keywords:** *Camellia sinensis*, preservation storage, postharvest metabolism, postharvest gene expression

## Abstract

Proper postharvest storage preserves horticultural products, including tea, until they can be processed. However, few studies have focused on the physiology of ripening and senescence during postharvest storage, which affects the flavor and quality of tea. In this study, physiological and biochemical indexes of the leaves of tea cultivar ‘Yinghong 9′ preserved at a low temperature and high relative humidity (15–18 °C and 85–95%, PTL) were compared to those of leaves stored at ambient conditions (24 ± 2 °C and relative humidity of 65% ± 5%, UTL). Water content, chromatism, chlorophyll fluorescence, and key metabolites (caffeine, theanine, and catechins) were analyzed over a period of 24 h, and volatilized compounds were determined after 24 h. In addition, the expression of key biosynthesis genes for catechin, caffeine, theanine, and terpene were quantified. The results showed that water content, chromatism, and chlorophyll fluorescence of preserved leaves were more similar to fresh tea leaves than unpreserved tea leaves. After 24 h, the content of aroma volatiles and caffeine significantly increased, while theanine decreased in both groups. Multiple catechin monomers showed distinct changes within 24 h, and EGCG was significantly higher in preserved tea. The expression levels of *CsFAS* and *CsTSI* were consistent with the content of farnesene and theanine, respectively, but *TCS1* and *TCS2* expression did not correlate with caffeine content. Principal component analysis considered results from multiple indexes and suggested that the freshness of PTL was superior to that of UTL. Taken together, preservation conditions in postharvest storage caused a series of physiological and metabolic variations of tea leaves, which were different from those of unpreserved tea leaves. Comprehensive evaluation showed that the preservation conditions used in this study were effective at maintaining the freshness of tea leaves for 2–6 h. This study illustrates the metabolic changes that occur in postharvest tea leaves, which will provide a foundation for improvements to postharvest practices for tea leaves.

## 1. Introduction

Tea from *Camellia sinensis* is among the world’s three most popular beverages. Tea is rich in polyphenols, caffeine, theanine, and terpenoids, which collectively provide a wealth of benefits to humans and contribute to its unique taste and aroma [1]. Anti-aging, cancer prevention, and other health benefits continue to be uncovered, boosting tea consumption and demand [2]. With the rapid growth of the global tea industry and the increasing number of consumers, the world’s per capita tea consumption has increased by an average of 2.8% annually in the past decade (Food and Agriculture Organization of the United Nations, https://www.fao.org/international-tea-day/en/, accessed on 5 December 2021).

Tea is a processed food produced from the tender leaves of *Camellia sinensis*. The chemical components of fresh tea leaves are essential for their market value. Nonvolatile components are generally responsible for taste, while volatile components affect aroma [3]. Similar to fruits, vegetables, and flowers, plucking tea leaves does not mean the end of life, but the beginning of post-ripening and senescence. As with other agricultural products, proper postharvest storage is critical for tea [4]. Oxidization reactions change the overall phytochemical composition of tea leaves and alter the organoleptic profile of the final tea product, decreasing its commercial value [5]. Practices adopted by enterprises typically include chilled temperature storage to preserve sensory and quality components [5].

The environment, which includes temperature and relative humidity, affects leaf metabolism during the postharvest period [6]. Many postharvest approaches have been implemented to preserve the freshness of tea leaves, with low temperature storage being the major approach to slow down metabolic activities [5]. Studies have found that tea leaves have reduced aroma content after plucking, and immediate storage at low temperature can enrich aromatic compounds [7,8]. However, compared to the extensive research on the compositional changes during tea processing, few studies have focused on the complex physiology of tea leaves during postharvest storage [9,10]. Physiological indexes and changes in tea metabolites during storage would show the influence of postharvest storage conditions on tea quality. Therefore, to improve the process of postharvest storage, it is important to understand the physiology of ripening and senescing leaves, including metabolite fluctuation and the expression of biosynthesis genes.

Although previous studies have investigated the metabolism of volatiles and other constituents, including theanine, caffeine, and catechins in postharvest tea leaves [5,8,11], few studies have integrated metabolic phenotypes and the gene expression analysis of postharvest leaves of *Camellia sinensis*. Enlightened by the preservation measures of other studies [8], fresh leaves were divided into two groups: preserved tea leaves (PTL) and unpreserved tea leaves (UTL). Physiological and biochemical indexes were compared among the two groups. In addition, genes associated with the biosynthesis of major metabolites were quantified. This study is a step towards clarifying the physiology of postharvest tea leaves and evaluating the effect of preservation on tea quality, which will help producers achieve greater economic benefits.

## 2. Results

### 2.1. Changes in Water Content, Color, and Degree of Damage during Storage

Suitable storage conditions play an important role in preserving freshly plucked tea leaves, and water content, color change, and degree of damage are important indexes to evaluate preservation. To explore the time-dependent changes in PTL and UTL, water content, color values (L*, a*, and b*), and chlorophyll fluorescence (Fv/Fm) were measured at 0, 1, 2, 4, 6, 12, and 24 h (Figure 1 and Appendix A).

Throughout the 24 h monitoring period, PTL maintained a water content level closer to that of FTL than UTL, and the difference between UTL and PTL was significantly different at 24 h (*p* < 0.05). The water content of FTL was 78.49% (Figure 1). After 24 h, the water content of UTL decreased to 75.77%, and PTL fluctuated between 77.15% to 78.57% water.

During storage, UTL gradually lost their luster and their color changed relative to FTL, as indicated by the color difference values (L*, a*, and b*, Appendix A). At 2 h postharvest, the color of UTL and PTL had not changed relative to FTL (i.e., ΔE < 2); however, significant color differences were observed in UTL after 4 h (∆E > 3.5). On the other hand, PTL maintained an ∆E < 3 throughout the 24 h monitoring period, indicating that the color difference would only be noticeable by an expert (Figure 1B).

Fv/Fm values (Figure 1C), calculated from chlorophyll fluorescence imaging (Figure 1D), showed that the degree of damage to UTL was much higher than the damage observed in PTL, as indicated by their lower Fv/Fm values. At 4 h, the Fv/Fm values significantly decreased in UTL, while the value changed only slightly in PTL. After 24 h, the Fv/Fm value of UTL changed to 0.630, a 13.0% decrease relative to FTL, while that of PTL dropped to 0.683, or 5.6% less than FTL.

### 2.2. Changes in Caffeine, Theanine, and Catechins Contents during Storage

Among the numerous secondary metabolites in tea, caffeine, theanine, and catechins are highly related to tea’s pleasant flavors [12]. Thus, dynamic changes to their contents were measured, via HPLC-UV/Vis spectroscopy, to evaluate the potential effects of storage conditions on tea flavor (Figure 2). Over the course of 24 h storage, the levels of caffeine in UTL changed greatly, while that in PTL changed relatively gradually (Figure 2A). As shown in Figure 2B, theanine content in both groups decreased markedly relative to FTL, and the contents were significantly different between the two groups (*p* < 0.05).

From the dynamic changes of catechins, we found that the contents of C and EC in PTL were substantially higher than in UTL in the first 6 h, but lower after 12 h (Figure 2C,D). In both groups, the contents of GC and EGC decreased after 24 h (Figure 2E,F), while the content of GCG significantly increased after 24 h (Figure 2G). ECG and EGCG were the major monomeric catechins in tea samples. By 24 h postharvest, ECG had reached 60 mg/g (dw) in both PTL and UTL, while EGCG was 63.18 mg/g (dw) and 54.01 mg/g (dw), respectively, in PTL and UTL (Figure 2H,I). The content of ECG significantly increased within 24 h in both groups, while the content of EGCG was substantially higher in PTL than in UTL after 24 h. Overall, each metabolite displayed a different trend over the course of the 24 h storage period that largely depended on whether they were preserved or not.

### 2.3. Comprehensive Analysis of Volatilized Compounds in UTL and PTL

HS-SPME/GC-MS analysis revealed a total of 52 volatile compounds in FTL, PTL, and UTL (Table 1) that were composed of alcohols (10), aldehydes (5), ketones (5), alkenes and terpenes (10), acids (1), alkanes (8), esters and lactones (10), and others (3). A total of 52, 37, and 34 compounds were identified in FTL, UTL, and PTL, respectively. The total content of volatilized compounds in UTL increased by 11.8%, while the total in PTL increased by 28.4% (Appendix A). As illustrated in Figure 3A, there were apparent differences in individual volatiles between the two groups. Compared with UTL, the aroma volatile content in PTL increased to a larger extent, including for 1-octen-3-ol, cis-β-farnesene, and trans-β-ionone. In addition, compounds known for their sweet, fruity, and floral odor, especially cis-β-farnesene, were more than 11.3 times higher in PTL than in UTL.

Because of the large number of volatiles, we focused on 16 major volatiles that play key roles in tea fragrance [7,13,14,15,16,17,18,19] (Figure 3B). There was no significant change in the contents of ten aroma volatiles, namely, β-myrcene, D-limonene, terpineol, geraniol, phytol, neophytadiene, linalool oxide, methyl salicylate, methyl hexadecanoate, and dibutyl phthalate, between PTL and UTL. In contrast, there were significant differences (*p* < 0.05) in the contents of linalool and decanal between UTL and PTL, and the differences were even greater for β-ocimene, α-farnesene, nerolidol, and cedrol. α-Farnesene and nerolidol were not detected in PTL, while cedrol was 5 times higher in PTL than in UTL. The results indicated that 1-octen-3-ol, β-ocimene, cis-β-farnesene, trans-β-ionone, and cedrol increased in content under preserved storage conditions.

### 2.4. Expression Levels of Metabolite Biosynthesis Genes during Storage

To further explore the mechanisms behind observed metabolic changes, biosynthesis genes of terpene aroma, caffeine, theanine, and catechins were analyzed by qRT-PCR at different time points in UTL and PTL (Figure 4). The terpene aroma biosynthesis gene, farnesene synthase (*FAS*) was significantly upregulated in PTL within 24 h, but less so in UTL. In contrast, limonene synthase (*LMS*) was steadily downregulated during storage in PTL. The expression levels of germacrene D synthase (*GDS*) and 4-hydroxy-3-methylbutenlyl diphosphate reductase (*HDR*) had no apparent change. Mevalonate-5-pyrophosphate decarboxylase (*MVD*), terpene synthase 78 (*TPS78*), and terpene synthase 77 (*TPS77*) had lower expression than FTL in both groups.

The expression levels of tea caffeine synthase 1 (*TCS1*) and tea caffeine synthase 2 (*TCS2*) were upregulated at 4 h in PTL, but finally, they decreased at 24 h in both groups.

Key genes involved in theanine biosynthesis, alanine decarboxylase (*AlaDC*), theanine synthetase (*TSI*), and glutamine synthetase 2 (*GS2*), were analyzed. *TSI* was highly expressed at 12 h in UTL and at 6 h in PTL. However, *AlaDC* had a higher expression level in UTL than in PTL. *GS2* was downregulated after 24 h and had a high expression level at 4 h in PTL.

Key catechin biosynthesis genes, flavonoid 3’,5’-hydroxylase (*F3*’*5*’*H*), flavonoid 3-hydroxylase (*F3H*), leucoanthocyanidin reductase (*LAR*), anthocyanidin reductase (*ANR*), and serine carboxypeptidase-like acyltransferase 7 (*SCPL1A7*) were consistently upregulated in the early hours of PTL, but the expression level decreased after 12 h. The expression level of *F3*’*5*’*H* and *F3H* fluctuated greatly in UTL within 24 h. *LAR*, *ANR,* and *SCPL1A7* were significantly downregulated in UTL and had the lowest expression level at 24 h.

### 2.5. A Comprehensive Evaluation of Quality in UTL and PTL

PCA comprehensively evaluates sample differences based on multiple sets of index data. Physiological and biochemical property indexes for water, caffeine, theanine, and catechin content, which directly reflect the quality of tea leaves, were included in a PCA. The PC scores of the first two PCs were plotted (Figure 5): the first principal component (PC1) explained 35.3% of the total variation, and the second principal component (PC2) 21.7%. The obtained PC scores indicated the difference between samples by variance: each circle represented a sample, and the distance between circles indicated the difference between and within groups. The closer the circles, the higher the similarity. When considering PTL and UTL at the same time point, PTL was closer to FTL than UTL from 2 h to 6 h. The distance between PTL and FTL was much shorter than that between UTL and FTL after 2 h, whereas the distance difference was not apparent after 12 h, indicating that the quality of PTL was most similar to FTL in the 2 h to 6 h period.

## 3. Materials and Methods

### 3.1. Plant Materials and Sampling

The elite tea plant cultivar used in the present study, ‘Yinghong 9′, was planted in Yingde, Guangdong province, China (24.20° N, 113.40° E). One bud and two leaves without red stain or charred edges were plucked in August 2021. Eight kilograms of fresh tea leaves were stored in a temperature and humidity control chamber (Yishi Technology Co., Ltd., Hangzhou, China) at 15–18 ℃ and 85–95% relative humidity (PTL), and another 8 kg of fresh tea leaves were placed at ambient conditions, at 24 ± 2 °C with a relative humidity of 65 ± 5% (UTL). Each treatment (PTL and UTL) was sampled in triplicate at 0, 1, 2, 4, 6, 12, and 24 h after plucking, and placed at −80 °C until the analysis of flavor compounds and gene expression. Fresh tea leaves (FTL) served as the positive control.

### 3.2. Measurement of Water Content

The water content of FTL was detected using the suggested protocol from the Chinese National Standard GB/T8304-2013 (General Administration of Quality Supervision, Inspection, and Quarantine of the People’s Republic of China, 2013) [21] with minor changes. Briefly, a clean aluminum box was dried at 103 °C ± 2 °C with the cover slanted on the edge of the box for 1 h, cooled in a desiccator to room temperature, and weighed (m_1_, accurate to 0.0001 g). Then, approximately 4 g of FTL (m_0_) were placed in the pre-dried box and put in the desiccator at 120 °C for 2 h with the cover slanted on the edge of the box. After capping the box, samples were cooled in the desiccator to room temperature and weighed (m_2_, accurate to 0.0001 g). Finally, the sample was baked for another hour and weighed (m_3_, accurate to 0.0001 g) until the difference in weight between m_2_ and m_3_ did not exceed 0.0050 g. All samples were weighed on an electronic scale (BSA224S-CW, Sartorius). The water content of the samples was calculated as: water content (%) = (m_2_−m_1_)/m_0_ × 100%.

### 3.3. Measurement of Color Difference

The color of the tea leaves was determined using a chromameter CR-400 (Konica Minolta, Tokyo, Japan). The color was measured according to the international commission on an illumination color solid scale (CIE: L*a*b*): L* indicates lightness, a* stands for red (+), and green (−), and b* indicates yellow (+) and blue (−) [22,23]. The total color difference (ΔE) was calculated as follows:$$\Delta E=\sqrt{{\left(L_{\mathrm{ss}}-L_{\mathrm{ts}}\right)}^{2}+{\left(a_{\mathrm{ss}}-a_{\mathrm{ts}}\right)}^{2}+{\left(b_{\mathrm{ss}}-b_{\mathrm{ts}}\right)}^{2}}$$
where L_ss_, a_ss_, and b_ss_ represent the standard sample (FLT), and L_ts_, a_ts,_ and b_ts_ represent the test sample. Color difference was classified by the following scale: when ΔE < 1, the color difference was not noticeable; when 1 < ΔE < 2, the color difference was only noticeable by experienced observers; when 2 < ΔE < 3.5, the color difference was noticeable by inexperienced observers; and when 3.5 < ΔE < 5, the color difference was pronounced [24].

### 3.4. Measurement of Chlorophyll Fluorescence (Fv/Fm)

The maximum photochemical efficiency of PSII (Fv/Fm) was measured with an IMAGING-PAM chlorophyll fluorescence system (Heinz Walz GmbH, Effeltrich, Germany) using default parameters according to Yu’s [25]. After tea leaves adapted to the dark for 20 min, Fo was measured. Then, a one-second saturation pulse occurred that completely closed all PSII receptors, and Fm was measured. The maximum photosynthetic efficiency (Fv/Fm) was calculated as:maximum photosynthetic efficiency (Fv/Fm) = (Fm − Fo)/Fm

### 3.5. Quantification of Caffeine Contents

Caffeine standards were purchased from Beijing Weiye Research Institute of Metrology and Technology (Beijing, China). Caffeine content was detected by HPLC-UV/Vis spectrometry according to the Chinese National Standard (GB/T 8312-2013) [26] with changes. Freeze-dried tea powder (0.1 g) was extracted with 30 mL 1.5% magnesium oxide in ultrapure water (*w*/*v*) at 100 °C for 30 min. One mL of the liquid supernatant was filtered through a 0.22 mm Millipore membrane, and 10 μL of the filtrate was injected into an XSelect HSS C18 SB column (4.6 × 250 mm, 5 mm, Waters Technologies, Milford, MA, USA) at a flow rate of 0.9 mL/min, with the column at 35 ± 1 °C. Caffeine was detected at 280 nm on a Waters Alliance E2695 equipped with a 2489 UV/Vis detector (Waters Technologies, Milford, MA, USA). The mobile phases consisted of 100% methanol (A) and 100% ultrapure water (B). Compounds were eluted under isocratic conditions: 30% A and 70% B.

### 3.6. Quantification of Theanine Contents

Theanine standards were purchased from Shanghai Yuanye Bio-Technology Co., Ltd. (Shanghai, China). Theanine content was detected by HPLC-UV/Vis spectroscopy according to the Chinese National Standard (GB/T 23193-2017) [27] with a Waters Alliance E2695 equipped with a 2489 UV/Vis detector (Waters Technologies, Milford, MA, USA). Fine freeze-dried tea powder (0.1 g) was extracted with 10 mL ultrapure water at 100 °C for 30 min. One mL of the liquid supernatant was filtered through a 0.22 mm Millipore membrane, and 10 μL of each filtrate was injected onto an RP-C18 column (250 mm × 4.0 mm, 5 μm) maintained at 35 ± 1 °C. The mobile phases consisted of 100% ultrapure water (A) and 100% acetonitrile (B). The flow rate was 0.5 mL/min, and the HPLC program was as follows: 100% solvent B from 0–12 min, 100% B to 20% B from 12–14 min, 20% B from 14–19 min, 20% B to 100% B from 19–20 min, and 100% B from 20–25 min. Theanine was detected at 210 nm.

### 3.7. Quantification of Catechin Contents

Catechin (C), epicatechin (EC), gallocatechin (GC), epigallocatechin (EGC), epicatechin gallate (ECG), gallocatechin gallate (GCG), and epigallocatechin gallate (EGCG) standards were purchased from Shanghai Yuanye Bio-Technology Co., Ltd. (Shanghai, China). C, EC, GC, EGC, ECG, GCG, and EGCG contents were detected by HPLC-UV/Vis spectroscopy based on GB/T 8313-2018 [28,29]. Briefly, 0.2 g fine freeze-dried tea powders were extracted with 8 mL 70% methanol. One mL of the supernatant was filtered through a 0.22 mm Millipore membrane, and the filtrate was injected into an XSelect HSS C18 SB column (4.6 × 250 mm, 5 mm, Waters Technologies, Milford, MA, USA). Catechin monomers were eluted with 0.1% aqueous formic acid (A) and 100% acetonitrile (B) as the mobile phases, using a gradient elution program. For the first five minutes, the mobile phase was 8% B; then from 5 min to 14 min, B was increased from 8–25%; finally, B was decreased from 25–8% from 14–30 min. Catechins were detected at 280 nm.

### 3.8. Analysis of Microextraction Compounds

Head-space solid-phase micro extraction/gas chromatography–mass spectrometry (HS-SPME/GC-MS) was used to analyze volatile compounds [17]. Briefly, 0.2 g FTL powders spiked with 10 µL internal standard solution (8.64 µg/mL ethyl decanoate in dichloromethane) was added to a 2 mL NaCl saturated solution in a 15 mL head-space vial (Agilent, MA, USA). The head-space vials were sealed with seal caps, and tin foil paper tied with adhesive tape was placed over the sealed cap. A divinylbenzene/carboxen/polydimethylsiloxane (DVB/CAR/PDMS) fiber (50/30 µm inner diameter, 2 cm length) (Supelco, Darmstadt, Germany) was inserted into the head-space vial containing the sample for 40 min at 80 °C. After microextraction, the fiber was kept in the GC port for desorption for 3 min. An Agilent GC-MS 1890B-5977A (Agilent, Santa Clara, CA, USA) was employed for volatiles analysis. The HP-5MS chromatographic column (30 m × 0.25 mm × 0.25 µm) was loaded with high purity helium at a flow rate of 1.0 mL/min. The inlet and interface temperatures were 250 °C; the oven temperature was maintained at 50 °C for 1 min and then increased to 220 °C at a rate of 5 °C/min for 5 min. Ion source electron energy and temperature were 70 ev and 230 °C, respectively. Mass spectra were acquired in splitless mode within the mass range of 30–400 amu. Volatile compounds were identified based on their retention indices (RI) and similarity to spectra within the NIST 14 database. The retention times of standard saturated C9-C29 n-alkanes were analyzed under the same conditions to calculate RI. Volatile compounds with mass spectral match factors over 75 and differences between RI and RIs less than 30 were deemed acceptable.

### 3.9. Analysis of Gene Expression

Total RNA was extracted using a HiPure Plant RNA Mini Kit B (Magen, Guangzhou, China) according to the manufacturer’s instructions. cDNA synthesis was performed by reverse transcription of screened RNA in accordance with the protocol for the HiScript^®^ III RT SuperMix for qPCR (+gDNA wiper) (Vazyme, Nanjing, China). qRT-PCR was performed with a BioRad CFX384^TM^ Real-Time System (Bio-Rad, Hercules, CA, USA) under the following operating conditions: 95 °C for 5 min, 40 cycles of 95 °C for 10 s, 55 °C for 10 s, and 72 °C for 30 s. Actin was used as the internal reference, and the relative expression levels were calculated using the 2^−^^△△CT^ method [30]. Primers used in this study are listed in Appendix A.

### 3.10. Statistical Analysis

Excel 2019 (Microsoft, Washington, DC, USA) was used to process the data. Heat maps and principal component analysis (PCA) were generated with GraphPad Prism 9.0 (GraphPad Software, San Diego, CA, USA). Dunnett’s multiple comparisons and *t*-test, also calculated by GraphPad Prism 9.0, were used to analyze the statistics, and differences were considered statistically significant when *p* < 0.05 (*).

## 4. Discussion

In green tea production, which does not include fermentation, the optimal choice is to process fresh tea leaves immediately postharvest [31]. However, many fresh tea leaves are picked simultaneously in spring, and it is not feasible to process them in time before quality begins to deteriorate [32]. This poses a problem for producers: how to store postharvest fresh leaves to maintain leaf quality? Therefore, it is necessary to determine best practices for storing fresh leaves postharvest to ameliorate the effects of processing delay on tea quality [5]. Low temperature and high relative humidity are key factors that affect the quality of horticultural products. In this study, postharvest tea leaves were stored for 24 h under preserved or unpreserved conditions to compare the preservation effect, which was assessed with physiological and biochemical indexes.

Fresh tea leaves adjust their physiological conditions to adapt to different storage environments by absorbing and releasing water; thus, water content is one of the essential factors affecting the postharvest preservation of tea leaves. In this study, we found that PTL maintained a high level of water, which was close to the initial stage. The water contents of UTL decreased more than PTL after 24 h (75.77% vs. 77.15%, respectively), and the difference between the two groups was significant (*p* < 0.05). A previous study showed that, after 6 h, the water content of tea leaves decreased to 63.8–68.1% [17]. The relatively high water content in UTL from this study might have been caused by the storage method: we kept fresh leaves in a basket without spreading, which probably slowed down water loss. Furthermore, the atmospheric humidity during storage of the UTL was somewhat high (60%~70%), which may have inhibited the transpiration rate of the tea leaves during the 24 h storage period [33,34]. Even though the water content showed little overall change in the two groups, low temperature (15–18 °C) and high humidity (85–95%) for PTL had a substantially better preservation effect, as indicated by ∆E and Fv/Fm (Figure 1B,C).

Caffeine, theanine, catechins, and aroma volatiles are essential for tea quality and market value; their contents determine the color, freshness, strength, and aroma of tea [35]. Caffeine, responsible for the refreshed feelings tea can bring, is the main tea alkaloid accounting for 2–4% of tea dry weight [20]. Theanine contributes sweet and savory tastes, while catechins are responsible for the color, bitterness, and astringency of tea [2]. In a previous study, harvested tea leaves underwent a processing delay of 6, 12, 18, or 24 h at temperatures of 0, 5, and 25 °C to investigate the effect of postharvest processing and storage of Japanese-style green tea [5]. Analyzed green tea constituents included theanine, caffeine, and catechins. These metabolites displayed a substantial decrease when tea was stored at 25 °C postharvest. However, in this study, it seemed the decreased level of these metabolites was less than that measured in previous studies. The disparity may be attributed to the specific tea cultivar, or to conditions other than temperature that differed (e.g., the relative humidity). Aromatic components, such as 1-octen-3-OL, β-ocimene, cis-β-farnesene, and cedrol, significantly increased in PTL, consistent with previous studies employing low temperature conditions [8,36]. The patterns of changes to caffeine, theanine, and catechin content were distinctly different between PTL and UTL over the 24 h period.

To further understand the molecular mechanisms behind these metabolic differences, qRT-PCR was used to detect changes in the expression of CBGs, FBGs, TBGs, and TPSGs during storage. Interestingly, the expression levels of some genes were consistent with the measured contents of metabolites, such as *FAS*, which was correlative to the β-farnesene content, and *TSI*, which had the strongest correlation to theanine content. However, the expression levels of *TCS1* and *TCS2* were inconsistent with the caffeine content. We speculate that the high caffeine content was due to the transformation of other tea alkaloids into caffeine instead of de novo biosynthesis during preservation, but this needs to be further addressed.

The physiological and biochemical indexes of tea cultivar ‘Yinghong 9′ were used to evaluate the ability of preservation under low temperatures and high humidity to maintain the quality of tea leaves postharvest. Through a comprehensive assessment by PCA, we found that the preservation effect of PTL took effect from 2 h to 6 h. However, future studies using prolonged storage time and measuring a larger number of substances and genes are necessary to further explore the induction mechanism of low-temperature preservation.

## 5. Conclusions

In summary, our study combined metabolic phenotypes and gene expression to analyze physiological changes and evaluate the effects of postharvest tea leaf preservation. The inhibition of water evaporation in PTL did not lead to a sudden drop in water content, and ∆E and Fv/Fm data during the period of storage showed that the PTL had a lower degree of damage than UTL. The contents of secondary metabolites varied significantly according to storage mode and time. Volatile aroma components 1-octen-3-ol, β-ocimene, cis-β-farnesene, and trans-β-ionone were the main increased components in PTL, and it was shown through comprehensive judgment that PTL maintained a preferable cedrol. Moreover, the biosynthetic genes were linked to the synthesis of metabolites and terpenes, so at the molecular level, we found that *CsFAS* in TPSGs and *CsTSI* in TBGs showed the same trend as farnesene and theanine, indicating that the synthesis of farnesene and theanine were regulated by the above genes, respectively. Prime freshness was found during the period of 2 h to 6 h, which provides a storage time scheme for preserving fresh tea leaves in short time for production.

## Figures and Tables

**Figure 1 molecules-27-01708-f001:**
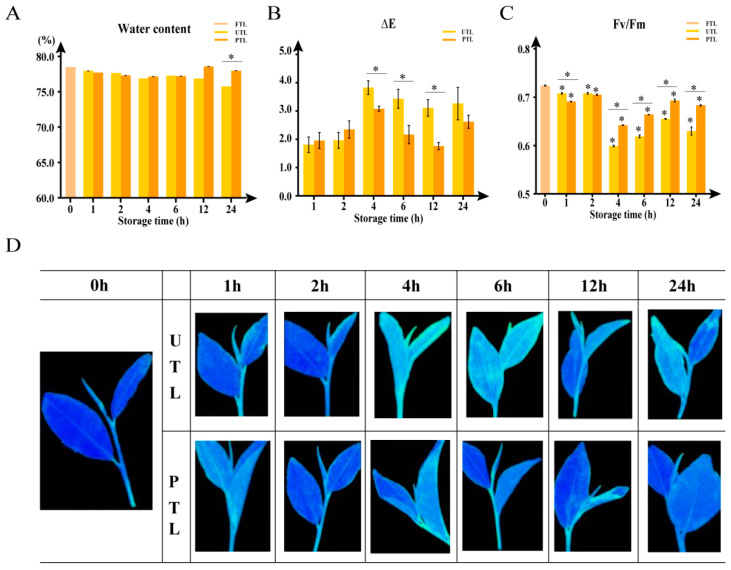
The effects of storage conditions on tea leaves, as indicated by changes in water content, color (∆E), and leaf damage (Fv/Fm) within 24 h. (**A**) The water content of FTL, UTL, and PTL at different time points. (**B**) ∆E of UTL and PTL within 24 h, relative to FTL. (**C**) The Fv/Fm values of FTL, UTL, and PTL over 24 h. (**D**) Images of chlorophyll fluorescence in FTL, UTL, and PTL at different time points. Dunnett’s multiple comparisons and *t*-test were used to identify significant differences (*, *p* < 0.05). Values are the mean ± SEM of all replicates.

**Figure 2 molecules-27-01708-f002:**
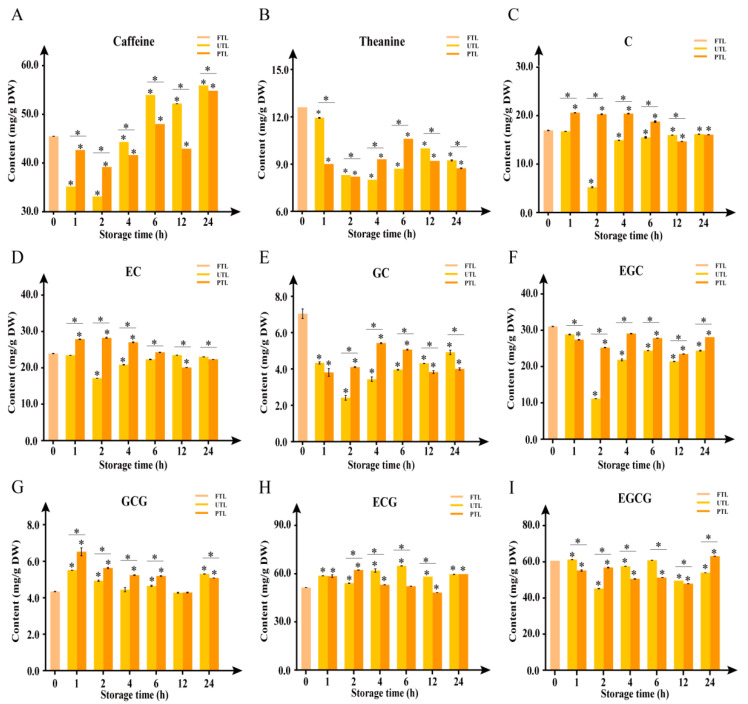
Changes in caffeine (**A**), theanine (**B**), and catechin (**C**–**I**) contents in preserved (PTL) and unpreserved (UTL) up to 24 h postharvest. C, catechin; EC, epicatechin; GC, gallocatechin; EGC, epigallocatechin; ECG, epicatechin gallate; GCG, gallocatechin gallate; EGCG, epigallocatechin gallate. Dunnett’s multiple comparisons and *t*-test identified significant differences (*, *p* < 0.05). Values are the mean ± SEM of all replicates.

**Figure 3 molecules-27-01708-f003:**
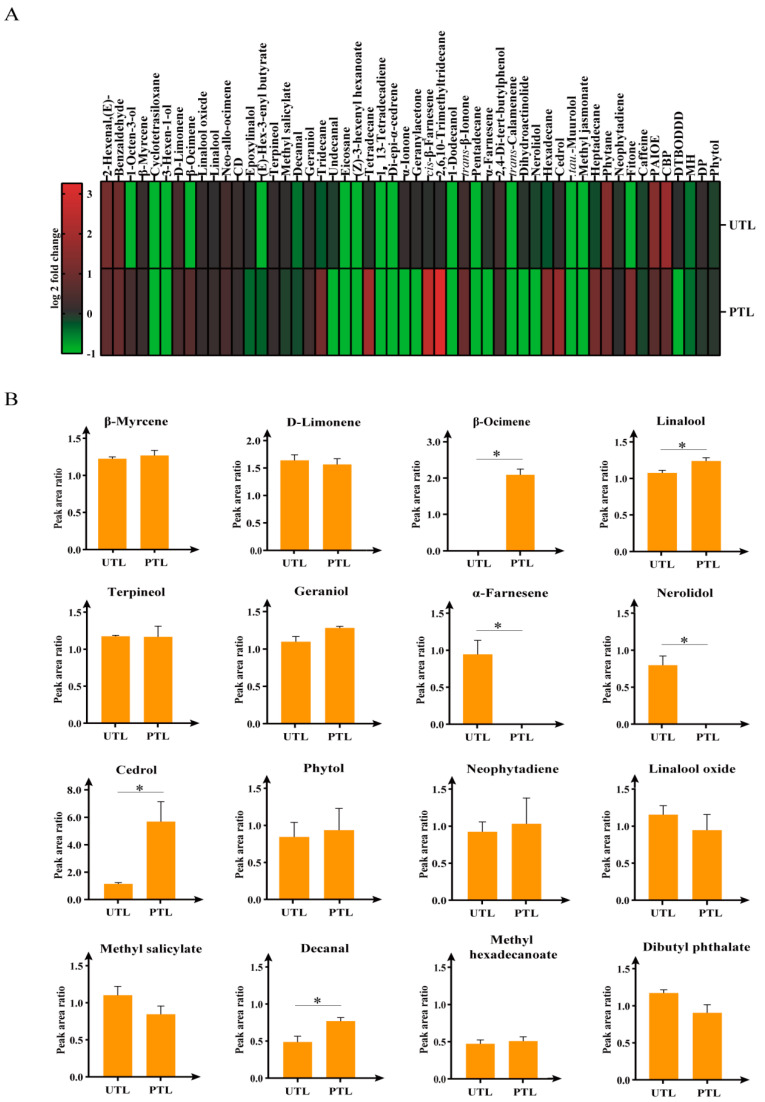
Changes in volatile compounds of tea leaves in UTL and PTL at 24 h, relative to FTL. (**A**) Heatmap showing changes of 52 volatile compounds. Red and green represent a positive and negative fold change, respectively, relative to FTL. (**B**) Changes of 16 key aroma compounds in UTL and PTL. Dunnett’s multiple comparisons and *t*-test were used to identify significant differences (*, *p* < 0.05). Values are the mean ± SEM of all replicates. CD, cyclopentasiloxane, decamethyl-; PAIOE, phthalic acid, isobutyl octyl ester; CBP, cyclohexyl butyl phthalate; DTBODDD, 7,9-Di-tert-butyl-1-oxaspiro (4,5) deca-6,9-diene-2,8-dione; MH, methyl hexadecanoate; DP, dibutyl phthalate.

**Figure 4 molecules-27-01708-f004:**
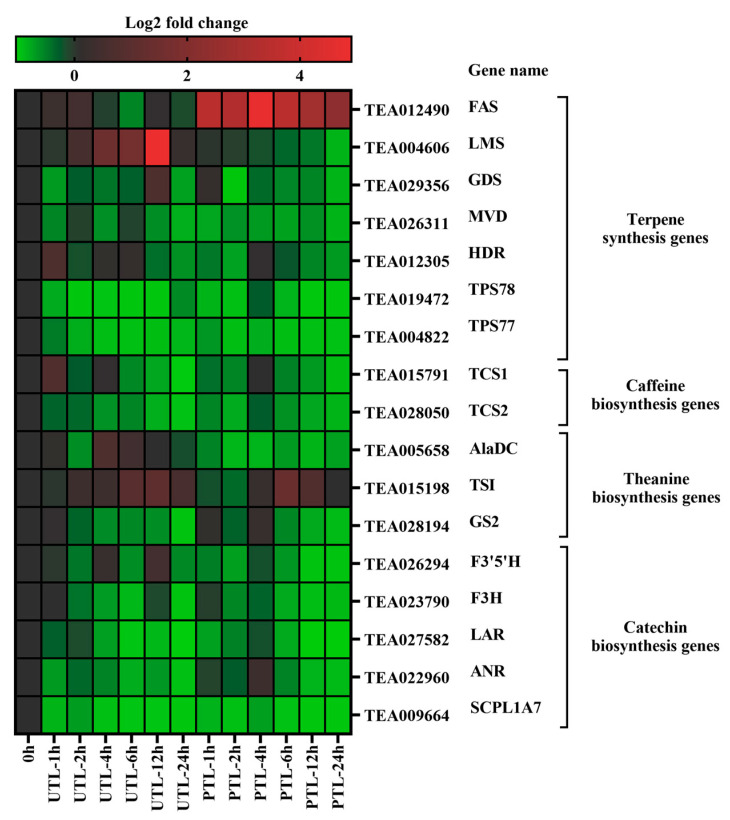
Heatmap for expression levels of biosynthetic genes in UTL and PTL relative to FTL. Red represents upregulation; green represents downregulation. *FAS*, farnesene synthase; *LMS*, limonene synthase; *GDS*, germacrene D synthase; *MVD*, mevalonate-5-pyrophosphate decarboxylase; *HDR*, 4-hydroxy-3-methylbutenlyl diphosphate reductase; *TPS78*, terpene synthase 78; *TPS77*, terpene synthase 77; *TCS1*, tea caffeine synthase 1; *TCS2*, tea caffeine synthase 2; *AlaDC*, alanine decarboxylase; *TSI*, theanine synthetase; *GS2*, glutamine synthetase 2; *F3’5’H*, flavonoid 3’,5’-hydroxylase; *F3H*, flavonoid 3-hydroxylase; *LAR*, leucoanthocyanidin reductase; *ANR*, anthocyanidin reductase; *SCPL1A7*, serine carboxypeptidase-like acyltransferase 7.

**Figure 5 molecules-27-01708-f005:**
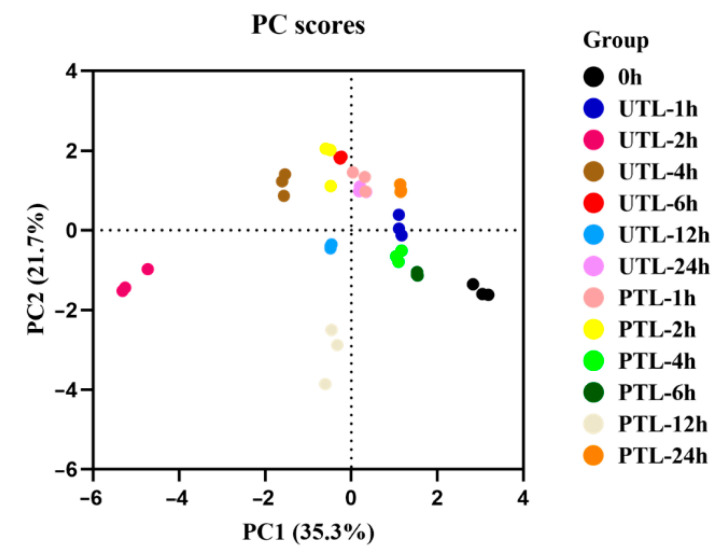
Graph of PC scores based on PCA.

**Table 1 molecules-27-01708-t001:** Volatile compounds in PTL and UTL.

					Relative Contents	
No.	Compound ^a^	RI ^b^	RT ^c^	Aroma Description ^d^	UTL	PTL
1	2-Hexenal, (E)- ^e^		854	Green, leafy, fruity	3.312 ± 0.507	2.423 ± 0.237
2	Benzaldehyde ^e^	967	962	Almond, burnt sugar	2.957 ± 0.398	2.706 ± 0.764
3	1-Octen-3-ol ^e^	981	982	Sweet, earthy, mushroom-like	n.d.	1.535 ± 0.176
4	β-Myrcene ^e^	994	993	Woody, resinous, musty	1.225 ± 0.024	1.269 ± 0.069
5	Cyclotetrasiloxane, octamethyl-	1000		—	n.d.	n.d.
6	3-Hexen-1-ol, acetate, (Z)- ^e^	1009		Grass	n.d.	n.d.
7	D-Limonene ^e^	1031	1030	Citrus, lemon, orange-like, green	1.639 ± 0.103	1.566 ±0.105
8	β-Ocimene ^e^	1039	1044	Sweet, herb	n.d.	2.092±0.159
9	Linalool oxide ^e^	1083		Flower	1.155 ± 0.122	1.181 ± 0.180
10	Linalool ^e^	1104	1100	Floral, sweet, grape-like, woody	1.076 ± 0.035	1.238 ± 0.046
11	Neo-allo-ocimene ^e^	1132	1131	Sweet, floral, nutty, herbal, peppery	1.492 ± 0.054	1.606 ± 0.089
12	Cyclopentasiloxane, decamethyl ^e^	1157		—	1.226 ± 0.038	1.177 ± 0.051
13	Epoxylinalol ^e^	1176	1183	Floral	1.055 ± 0.035	0.548 ± 0.028
14	(E)-Hex-3-enyl butyrate ^e^	1188	1185	Fruity, green, vanilla, cream	n.d.	0.612 ± 0.095
15	Terpineol ^e^	1194	1190	Pleasant, floral	1.173 ± 0.014	1.167 ± 0.143
16	Methyl salicylate ^e^	1197	1191	Minty, fresh, sweet	1.102 ± 0.120	0.845 ± 0.110
17	Decanal ^e^	1207	1200	Soap, orange peel, tallow	0.487 ± 0.079	0.768 ± 0.049
18	Geraniol ^e^	1231	1250	Rose-like, sweet, honey-like	1.133 ± 0.072	1.453 ± 0.021
19	Tridecane ^e^	1300		Alkane	0.666 ± 0.032	3.259 ± 0.312
20	Undecanal ^e^	1308	1308	Rose, waxy, oily	0.865 ± 0.150	n.d.
21	Eicosane ^e^	1326		Alkane	n.d.	n.d.
22	(Z)-3-hexenyl hexanoate ^e^	1383		Fruity, waxy, green, fatty, winey	n.d.	n.d.
23	Tetradecane ^e^	1400		Alkane	0.938 ± 0.072	6.995 ± 0.752
24	1,13-Tetradecadiene	1410		—	n.d.	n.d.
25	Di-epi-α-cedrene ^e^	1419		—	n.d.	n.d.
26	α-Ionone ^e^	1431	1433	Floral, violet-like, powdery, berry-like	1.153 ± 0.110	n.d.
27	Geranylacetone ^e^	1455	1454	Fresh floral, sweet-rosy	1.106 ± 0.042	n.d.
28	*cis*-β-Farnesene ^e^	1458	1457	Citrus, green	0.984 ± 0.096	11.089 ± 0.956
29	2,6,10- Trimethyltridecane	1462	1461	—	1.053 ± 0.097	18.202 ± 3.610
30	1-Dodecanol ^e^	1475	1480	Sweet, fatty	n.d.	n.d.
31	*trans*-β-Ionone ^e^	1490	1490	Violet-like, raspberry, floral	1.395 ± 0.137	2.891 ± 0.461
32	Pentadecane ^e^	1500		Alkane	n.d.	n.d.
33	α-Farnesene ^e^	1510	1508	Woody, green, floral, herbal	0.944 ± 0.228	n.d.
34	2,4-Di-tert-butylphenol	1515	1518	—	1.386 ± 0.123	1.579 ± 0.128
35	*trans*-Calamenene ^e^	1529	1529	Herbal, spicy	n.d.	n.d.
36	Dihydroactinolide ^e^	1537	1538	Musky, coumarin-like	0.941 ± 0.059	n.d.
37	Nerolidol ^e^	1567	1567	Wood, flower, wax	0.797 ± 0.124	n.d.
38	Hexadecane ^e^	1600		Alkane	0.704 ± 0.052	4.483 ± 0.112
39	Cedrol ^e^	1610	1609	Sweet, fruity, cedar-like	1.147 ± 0.090	5.691 ± 1.452
40	*.tau.*-Muurolol ^e^	1649	1648	Herb, weak spice	n.d.	n.d.
41	Methyl jasmonate ^e^	1653	1655.4	Jasmine	n.d.	n.d.
42	Heptadecane ^e^	1700		Alkane	0.764 ± 0.072	3.221 ± 0.309
43	Phytane	1809	1795	—	4.193 ± 0.358	3.035 ± 0.912
44	Neophytadiene ^e^	1840	1837	Fresh	0.924 ± 0.132	1.031 ± 0.349
45	Fitone ^e^	1847	1847	—	n.d.	3.681 ± 1.311
46	Caffeine	1854	1842	—	0.801 ± 0.087	0.808 ± 0.142
47	Phthalic acid, isobutyl octyl ester	1872		—	4.368 ± 0.364	2.104 ± 0.446
48	Cyclohexyl butyl phthalate ^e^	1919	1892	Mild	5.801 ± 0.552	2.391 ± 0.837
49	7,9-Di-tert-butyl-1-oxaspiro (4,5) deca-6,9-diene-2,8-dione	1924	1916.8	—	1.005 ± 0.081	n.d.
50	Methyl hexadecanoate ^e^	1928	1925	Oily, waxy, fatty	0.471 ± 0.052	0.508 ± 0.058
51	Dibutyl phthalate ^e^	1967	1969	Slight, aromatic	1.173 ± 0.043	0.904 ± 0.110
52	Phytol ^e^	2115	2116	Floral, balsam, powdery, waxy	0.844 ± 0.196	0.935 ± 0.295

^a^ Identification method: retention index in agreement with the literature value; mass spectrum comparison using the NIST 14 library. ^b^ Retention index was calculated based on the retention time of standard saturated C9-C29 n-alkanes under the same conditions. ^c^ The published retention index of compounds in NIST 14 library. ^d^ Aroma description found in references [14,18], the Flavornet database (https://www.flavornet.org/flavornet.html, accessed on 5 December 2021), and PubChem (https://pubchem.ncbi.nlm.nih.gov/, accessed on 5 December 2021). ^e^ Aroma volatile compounds identified from references [7,13,14,17,18,19,20], the Flavornet database (https://www.flavornet.org/flavornet.html, accessed on 5 December 2021), PubChem (https://pubchem.ncbi.nlm.nih.gov/, accessed on 5 December 2021), the Flavor Library (https://www.femaflavor.org/flavor-library, accessed on 5 December 2021), and Ichemistry (http://www.ichemistry.cn/, accessed on 5 December 2021). ‘—’, no aroma description information was found in the literature. ‘n.d.’, the compound was not detected.

## Data Availability

Not applicable.

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
