# Peer review of "The Physiology of Postharvest Tea (Camellia sinensis) Leaves, According to Metabolic Phenotypes and Gene Expression Analysis"

_molecules, 2022, doi:10.3390/molecules27051708_

Round 1

Reviewer 1 Report

There is many articles connected to tea topic. Therefore, Authors have t more express the importance of their project.

Abstract: This is not a conclusion that could end the article: "This study illustrates the metabolic changes that occur in postharvest tea leaves, which will provide a foundation for improvements to postharvest practices for tea leaves".

"low", "high", "ambient" it is not specific conditions, please try to be more specific.

Methodology: Please add the name and producer of scale that was used to measure the water content. Moreover, please add references to all mentioned methodologies.

The samples weren't closed with sealed caps? "The head-space vials were sealed with tin foil paper and adhesive tape.

Should be "microextraction". It cannot be said, "aroma volatile". Aroma-active compounds can be evaluated by molecular sensory concept with GC-olfactometry, and volatilized compounds can be determined using GC-MS or GC-MS/MS measurement.

How long the fiber was kept in GC-port?

"key aroma compounds" did the Authors mean GC peaks with the highest area peaks? Because according to obtained data the aroma-active regions of tea cannot be noticed.

Relative contents also can have units. There is no standard deviations and statistics.

I recommend to avoid usage of word "key" too much, and in my opinion the text requires extensive English corrections.

How it is possible to do distinguish volatile and aroma compounds?  In my opinion there is a serious misunderstanding in interpretation of volatile compounds determination.

There is no conclusion subchapter in this version of manuscript.

Author Response

There is many articles connected to tea topic. Therefore, Authors have t more express the importance of their project.

R1: Thank you very much for your comments. Many fresh tea leaves are picked simultaneously in spring, and it is not feasible to process them in time before quality begins to deteriorate. This poses a problem for producers: how to store postharvest fresh leaves to maintain leaf quality? Therefore, it is necessary to determine best practices for storing fresh leaves postharvest to ameliorate the effects of processing delay on tea quality. We have added more description about the importance of our project in the introduction section (Line 54, 64-66 and 76-78 highlight in yellow).

Abstract: This is not a conclusion that could end the article: "This study illustrates the metabolic changes that occur in postharvest tea leaves, which will provide a foundation for improvements to postharvest practices for tea leaves".

R2: Thank you for your comments. We revised the final summary of the abstract (Line 28-32 highlight in yellow).

"low", "high", "ambient" it is not specific conditions, please try to be more specific.

R3: Thank you for your comments. We have added specific conditions about "low", "high" and "ambient"(Line 16-17 highlight in yellow).

Methodology: Please add the name and producer of scale that was used to measure the water content. Moreover, please add references to all mentioned methodologies.

R4: Thank you for your comments. The name and producer of scale that was used to measure the water content (Line 251-252 highlight in yellow) and references have been added (Line 243, 269, 278, 290, 305 and 344 highlight in yellow).

The samples weren't closed with sealed caps? "The head-space vials were sealed with tin foil paper and adhesive tape.

R5: The samples closed with sealed caps and covered with a tin foil paper tied by adhesive tape. Thank you for your comments and the description have been corrected (Line 319-321 highlight in yellow).

Should be "microextraction". It cannot be said, "aroma volatile". Aroma-active compounds can be evaluated by molecular sensory concept with GC-olfactometry, and volatilized compounds can be determined using GC-MS or GC-MS/MS measurement.

R6: Thank you, we have corrected our description (Line 19, 134, 139, 314, 316, 325, 330 and 333 highlight in yellow).

How long the fiber was kept in GC-port?

R7: Thank you for your comments. The fiber was kept in GC-port for 3min and more details have been added in manuscript (Line 323-324 highlight in yellow).

"key aroma compounds" did the Authors mean GC peaks with the highest area peaks? Because according to obtained data the aroma-active regions of tea cannot be noticed.

R8: No, "key aroma compounds" were not the compounds with highest area peaks. "Key aroma compounds" were the aroma volatiles that play key roles in tea fragrance according to publishes. we have explained this and added the references in the manuscript (Line 147 highlight in yellow).

Relative contents also can have units. There is no standard deviations and statistics.

R9: Thank you for your comments. The standard deviations were added in the Table1. The units of relative contents is optional, please see the reference “Characterisation of odorant compounds and their biochemical formation in green tea with a low temperature storage process. We prefer to have no units, thank you for your understanding.

I recommend to avoid usage of word "key" too much, and in my opinion the text requires extensive English corrections.

R10: Thank you for your comments and the manuscript have been revised by native English speaker. Attached please see the certificate.

How it is possible to do distinguish volatile and aroma compounds?  In my opinion there is a serious misunderstanding in interpretation of volatile compounds determination.

R11: Thank you for your comments and the description have been modified in the full manuscript.

There is no conclusion subchapter in this version of manuscript.

R12: Thank you for your comments. We’ve added the conclusion subchapter after the Discussion section (Line 411-425 highlight in yellow).

Reviewer 2 Report

Manuscript Number: molecules-1607095

Title: The physiology of postharvest tea (Camellia sinensis) leaves, according to metabolic phenotypes and gene expression analysis

The paper assesses the metabolic changes during postharvest tea leaves storage. The paper is well written and analyze the physiology of the ripening and senescence which affect the flavor and quality of tea during postharvest storage. The experimental design is well performed and described, and the subject is relevant to the aim and scope of the journal. In my opinion the manuscript can be accepted for publication. Moreover, English language editing must be performed. Authors are invited to revise the manuscript according to the suggestions listed below:

  1. In the Materials and Methods section 2.10 (Statistical analysis) it is reported that the Dunnett’s multiple comparison was used to analyze the statistics. What software was used?
  2. Line 107: the authors must report the bibliographic reference for measurement of chlorophyll fluorescence
  3. Line 165: delete “per min”
  4. Line 189: the 2—ct method. What does it mean?
  5. Line 328: the authors, according PCA, must explained the total variance of the first two compoenents (PC1 x% and PC2 …). highlighting the higher contributions (loadings) on PC1 and PC2.

Author Response

1.In the Materials and Methods section 2.10 (Statistical analysis) it is reported that the Dunnetts multiple comparison was used to analyze the statistics. What software was used?

R1: Thank you very much for your comments. The sofeware we used to perform the Dunnett’s multiple comparison is GraphPad Prism 9.0 and we’ve added the information in the section of Statistical analysis (Line 349-350 highlight in yellow).

2.Line 107: the authors must report the bibliographic reference for measurement of chlorophyll fluorescence

R2: Thank you for your comments. We’ve reported the bibliographic reference for measurement of chlorophyll fluorescence in Line 269 highlight in yellow.

3.Line 165: delete “per min”

R3: Thank you for your comments, “per min” have been deleted accordingly (Line 328 highlight in yellow). 

4.Line 189: the 2—ct method. What does it mean?

R4: The ‘2-△△CT’ was the delta-delta CT value method used to calculate the relative gene expression according to the published. We have added the reference (Line 344 highlight in yellow).

5、Line 328: the authors, according PCA, must explained the total variance of the first two compoenents (PC1 x% and PC2 …). highlighting the higher contributions (loadings) on PC1 and PC2.

R5: Thank you very much for your comments. We have added concrete values that PC1 and PC2 explained as instructed (Line217-219 highlight in yellow). The values of PC1 and PC2 were also annotated in Fig.5.

Round 2

Reviewer 1 Report

Line 145: still "aroma volatile" expression was used

Line 308: should be microextraction

I have no further suggestions.

Author Response

Line 145: still "aroma volatile" expression was used

R1: Thank you very much for your comments. Changes have been made as instructed (Line 145).

Line 308: should be microextraction

R2: Thank you for your comments, but we could not find “microextraction” in Line 308. We speculated that ‘Line 308’ should be Line 323, and ‘extraction’ have been changed to ‘microextraction’ in Line 323 (Line 323 highlight in blue).
